# Long Survival of a Patient with Trisomy 18 and Dandy-Walker Syndrome

**DOI:** 10.3390/medicina55070352

**Published:** 2019-07-08

**Authors:** Leonardo Moura Ferreira de Souza, Augusto Galvão e Brito Medeiros, José Paulo Ribeiro Júnior, Aurea Nogueira de Melo, Sylvyo André Morais Medeiros Dias

**Affiliations:** 1Department of Pediatric of Federal, University of Rio Grande do Norte, Natal 58051-900, Brazil; 2Federal University of Rio Grande do Norte, Natal 58051-900, Brazil

**Keywords:** trisomy 18, Edwards syndrome, aneuploidy, Dandy-Walker, long-term survival

## Abstract

Trisomy 18 is a genetic disease resulting from an extra chromosome 18, characterized by a broad clinical spectrum, poor prognosis and low rates of survival. This is the case of a 12 year-old girl diagnosed with full trisomy 18, and multiple malformations, including Dandy-Walker Syndrome and congenital heart defects on long term survival. At nine months, a new echocardiogram showed a double outlet right ventricle, significant pulmonary stenosis, patent ductus arteriosus and ventricular septal defect. Cardiac surgery was performed at one year and seven months. Early surgical intervention and multidisciplinary follow-up may change the clinical outcome of the disease. Further studies are required to evaluate the benefit of invasive procedures such as cardiac surgery on survival of patients with trisomy 18.

## 1. Introduction

Trisomy 18 (T18), or Edwards syndrome (ES), is a genetic disease resulting from an additional 18 chromosome, characterized by a broad clinical spectrum, poor prognosis and low life expectancy. The live birth prevalence ranges from 1/3000 to 1/10,000 (average of 1 in 6000). It is the second most common autosomal trisomy, after 21 trisomy. Chromosome disturbance is usually associated with an abnormal chromosomal nondisjunction event during maternal gametogenesis [1]. It presents a high mortality rate, with approximately 90% of affected individuals dying within their first year of age [2,3]. Average survival ranges between 2.5 and 14.5 days after birth, although most deaths occur intrauterine [1]. Patients with trisomy 18 rarely reach adolescence [4,5]. There are more than 130 different abnormalities described in the literature associated with trisomy 18, which give rise to an heterogenous group of organ system anomalies, albeit none of them pathognomonic for the disease. Phenotypes frequently show neurological abnormalities, growth curve delay, internal organs malformation, and specific features in the face, thorax, abdomen, extremities, genitals, skin, nails and hair [1].

In this case, Dandy-Walker Syndrome (DWS) is concurrently reported. DWS is classically described as a partial or complete agenesis of the cerebellar vermis, cystic dilatation of the fourth ventricle and an expanded posterior fossa with upward displacement of the lateral sinus, tentorium and torcular herophili [6]. Such condition also confers a diverse cluster of clinical features, from asymptomatic cases to severe intellectual disability, and might be related to other systems abnormalities in up to half of the individuals [7].

In our review of the literature the association between DWS and T18 was described in just 27 cases. We therefore describe the case of a 12-year-old girl with ES and DWS, a rare association [8], that turns to be even more relevant when considering the age of the patient.

## 2. Case Report

A newborn girl was born after 36 weeks of gestation to a 39-year-old mother after her third pregnancy, which was complicated by gestational hypertension. The mother had no previous history of abortion and patient’s siblings were phenotypically normal. Parents were second-degree relatives and also phenotypically unremarkable. The patient had a family history of a second-degree cousin with Down Syndrome and an uncle with visual and motor deficiencies not specified. Her birth weight was 1980 g, with a length of 44 cm and the cephalic perimeter, not measured at first, was 30 cm on the 25th day of life. She was born with multiple malformations. Significant respiratory distress was found in the immediate postpartum and admission to a neonatal Intensive Care Unit (ICU) was followed, where she was given continuous positive airway pressure. Informed consent has been obtained from patient’s mother to reproduce images and other clinical data anonymously.

Following geneticist’s assessment, a set of clinical features was recognized, including syndromic facies, up-slanting palpebral fissures, microphthalmia, convergent strabismus on the left eye, short nose, micrognathia, thoracic asymmetry, severe scoliosis (Figure 1), polydactyly of right hand and left hand showing syndactyly between the 3rd and 4th fingers (Figure 2).

A G-band karyotype from peripheral blood was performed. Results (47, XX+18) demonstrated a female karyotype with complete trisomy of chromosome 18 (Figure 3), which led to the definitive diagnosis of ES.

In a subsequent cardiac evaluation, at nine months of age, the heart murmur was still present (grade 4/6), and a second echocardiogram was done, showing double outlet right ventricle, significant pulmonary stenosis, patent ductus arteriosus and ventricular septal defect. At this time, digoxin was prescribed and maintained until cardiac surgery, when the patient was one year and seven months. An echocardiogram after surgery revealed a good result, with a minimum interventricular communication and pulmonary insufficiency with mild hemodynamic repercussion.

On a closer evaluation of the genitourinary system, the patient exhibited recurrent urinary tract infections, besides a variety of abnormal results on imaging and functional studies (Figure 4). In a renal scintigraphy with dimercaptosuccinic acid, the presence of horseshoe kidneys and relative renal function of 69% in the right kidney and 31% in the left kidney were verified. An ultrasonography of the kidneys and urinary tract and a voiding cystourethrography demonstrated findings of mild hydronephrosis, hydroureter and vesicoureteral reflux grade II on the right side.

In regards to the gastrointestinal tract, she presented chronic constipation, with recurrent fecalomas formation, with repeated use of antacids for symptoms relieving. Such condition mostly characterizes neurogenic bowel dysfunction secondary to ES. In addition, she presented masticatory incapacity, necessitating food liquefaction, resulting in inadequate weight gain throughout her growth, which resulted in marked thinness (Z score <−3). Gastrostomy was proposed in an attempt to offer a greater nutritional contribution; however, her mother did not accept due to the inherent risks of the procedure.

A neurological examination showed global hypotonia, decreased strength, bilaterally exaggerated deep tendon reflexes and significant delay in neuropsychomotor development (NPMD). Around the age of two years, sporadic tonic-generalized seizures began and phenobarbital was prescribed since then. An electroencephalogram detected focal discharges. Computed tomography, without contrast of the head, showed the following findings: massive cystic formation in the posterior fossa inferiorly compressing the occipital parenchyma; hypoplasia of cerebellar vermis with calcifications in the left cerebellar hemisphere; moderate dilation of the IV ventricle; dilatation of the supratentorial ventricular system; elevated sinus and Torcular of Herophilus (Figure 5). These findings are compatible with DWS. Currently, the patient is being regularly followed by a pediatrics team in the regional university hospital.

## 3. Discussion

In recent decades, the mean survival of T18 patients is reported to be 10–14.5 days according to most population studies. The latest survival rates for <1 day, <28 days and <5 years are estimated, in percentage, as 78.1%, 37.2% and 12.3%, respectively [9]. A few cases of T18 children with long-term survival are reported in the literature, some occasionally reaching the second decade of life, although significant studies focusing on long-term survival are still lacking. Nevertheless, factors affecting mortality and long-term survival for T18 are being progressively reviewed, including type of mutation, populational variables, congenital conditions, physicians and family choices.

Improved long-term survival has been associated with partial or mosaic trisomies compared to children with full trisomy, albeit cytogenetic mosaics and Robertsonian translocations are considerably less prevalent across populations. In a British study, a 1-year survival rate of 70% was found for children with partial T18, in contrast to 8% for full trisomy [10]. It is noteworthy, however, that our patient reached 12 years of age despite her G-banded karyotype evidencing a full T18 diagnosis.

In a large American population-based cohort including 1113 children with T18, race, ethnicity and population density of maternal county of residence at delivery were analyzed. The study showed that higher survival is associated with non-Hispanic black mothers and metropolitan areas. In this same study, gestational age was found to be the strongest independent determinant of survival, especially as approximately one-half of the children were preterm. No significant association was linked to maternal age. In conformity with previous studies, gender was also pointed as an important risk factor for mortality, even after adjusted for other relevant variables, with 42.9% of 1-month survival probability for females and 24.6% for males, however the reason for this difference remains unclear [9]. Additionally, birth weight is also believed to influence survival rates, with very low birth weight accounting for worse outcomes, although this could be partly attributed to a tendency to withhold aggressive neonatal care for T18 children. As our patient had low birth weight (LBW) and was a mixed-race female, whose gestational age was close to 37 weeks, a better prognosis was expected at birth [11].

A diverse set of clinical features are frequently found in T18 patients, most organ systems being affected, from congenital heart defects (CHDs) to orofacial clefts, omphalocele, renal anomalies, and central nervous system malformations. A study analyzed mortality major CHDs and omphalocele. It was found that both conditions represent an increase in T18 mortality at 1 and 5 years, with the latter malformation accounting for a 50% higher risk [9]. It is notable that minor CHDs, such as patent ductus arteriosus, atrial septal defects, and ventricular septal defects, are not often related to reduced survival. In the discussed case, our patient had minor CHDs and defects not classified by these authors, which probably contributed to her longer survival.

In a retrospective cohort of 254 children with T18, it was observed that longer-term T18 survivors having CHDs or neurological malformations or other congenital anomalies in more organ systems, did not present with shorter survival times, and those individuals with longer survival had more admissions in their first year of life [12]. Reviewed data on 121 subjects with T18 who underwent CHD interventions in Canada and in the United States, finding that the most common surgical indication was heart failure in 36.8% of patients. Children with T18 had an in-hospital mortality rate of 13% and a 1-year survival after first surgery rate of approximately 70% [13]. It is remarkable that currently no studies have explored how quality-of-life factors influence decision making around procedural benefit for T18 patients [12].

Due to the high lethality of the disease, there are few data defining the impact of cardiac surgery on T18. Graham et al. (2004) obtained survival equivalent to 21 out of 24 patients undergoing cardiac surgery [14]. However, the need for a reassessment of each particular case to establish the real impact of the procedure was emphasized. In another study, surgical corrections were performed in patients aged 2 weeks to 41 months with 87.5% survival rate until survey completion [15]. That author suggests that cardiac surgery should be considered for these patients. Despite the small sample in the aforementioned studies, they suggest that patients may benefit from cardiac surgeries, thus contributing to the higher survival seen in the patient of the present report.

On the other hand, Janvier et al. (2016) concluded that the single most important factor associated with higher longevity in patients with full trisomy of 13 and 18 was the prenatal diagnosis [16]. In this study, 36% of children with a prenatal diagnosis lived <24 h and 47% were discharged home compared to 1% and 87%, respectively for children with a postnatal diagnosis. Children with a postnatal diagnosis were treated “as any other children” until the diagnosis, which may give them a survival advantage, independent of palliative care. This corroborates with the data that children submitted to intensive neonatal treatment present greater survival [17]. Other factors associated were with decreased survival were male gender, low-birth weight, and cardiac and/or cerebral anomaly [16]. Our patient received a postnatal diagnosis which may have been beneficial to her long survival.

Usually, patients with T18 are born underweight and have difficulty gaining weight later. There are specific growth curves in literature for these patients, since they tend to present a slow growth rate [1]. However, for the patient in question, curves used for normal children were used, which contributed to the fact that she was always considered to be below the expected weight.

Urological malformations are very common in patients with this syndrome. The most prevalent are: Horseshoe, polycystic, ectopic or hypoplastic kidneys, renal agenesis, hydronephrosis, hydroureter and duplicated ureters [1]. Several of the kidney changes described in the literature were compatible with those observed in this case.

Due to the rarity of the association between T18 and DWS, epidemiological data regarding the involvement of these two syndromes are scarce. In a review from three series of cases with total of 78 patients with DWS, 20 were also affected by T18 [8].

DWS is a rare central nervous system malformation affecting approximately 1 in 30,000 infants. It presents unknown etiology and is characterized by the presence of aplasia or hypoplasia of cerebellar vermis associated with cerebellar cyst connecting the posterior fossa and the fourth ventricle. Its diagnosis can be made by cranial image exams like magnetic resonance and computed tomography [8]. Hydrocephalus and cerebellar ataxia are usually associated with this clinical condition, but despite the cerebellar alterations, it was not possible to evaluate the presence of ataxic gait in the patient, as she was unable to ambulate.

The clinical findings of patients with DWM depend on the degree of nerve involvement and other neurological or systemic malformations that are present. Delayed NPMD is present in about 40% of cases of DWS. Hypotonia is also common, and there are documented cases of this syndrome whose main complaint was generalized hypotonia. Both findings also occur in T18 [1]. As these two neurological changes were observed in the patient in question, it is concluded that hypotonia and NPMD delay can be justified by both conditions.

## 4. Conclusions

Considering the severity, mortality and survival of patients with complete trisomy of chromosome 18, as well as the organic disorders present in this clinical case, an unfavorable outcome was expected, with death soon after birth. The authors believe that the early cardiac surgery and multidisciplinary follow-up consisting of geneticist, gastroenterologist, cardiologist, neurologist and nutritionist were the main agents responsible for the achieved clinical outcome, which allowed the patient to reach 12 years of age. More investigations need to be undertaken to help confirm the benefits of these measures. They also suggest the development of a clinical management protocol for patients with trisomy 18.

## Figures and Tables

**Figure 1 medicina-55-00352-f001:**
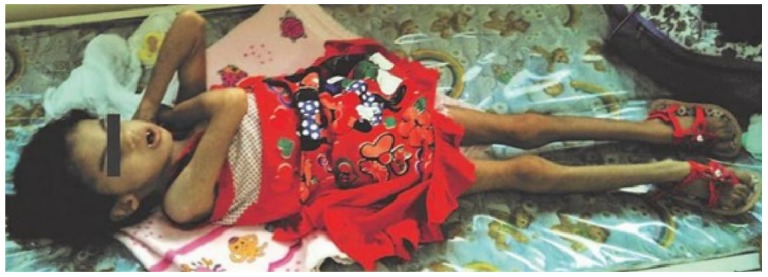
10-year-old girl presenting phenotypic changes from trisomy 18 and Dandy-Walker Syndrome, including low weight, poor bone structure and signs of cognitive delay.

**Figure 2 medicina-55-00352-f002:**
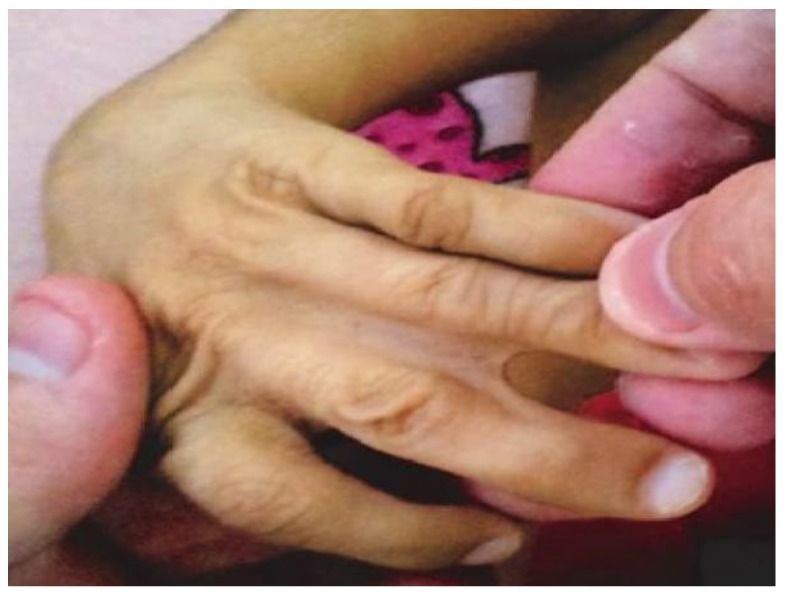
Syndactyly between the third and fourth left fingers in a 10-year-old girl with trisomy 18 and Dandy-Walker Syndrome.

**Figure 3 medicina-55-00352-f003:**
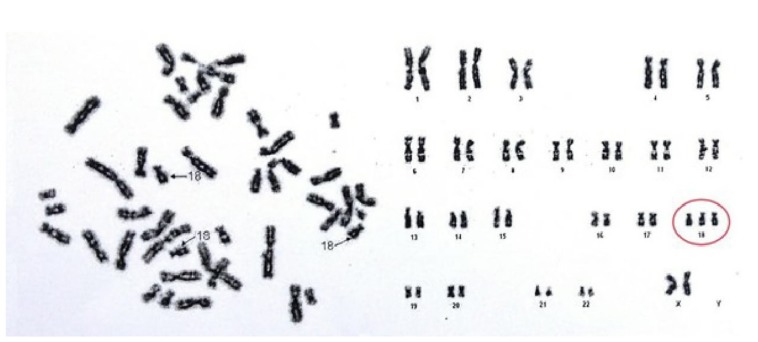
Patient’s karyotype, showing complete trisomy of chromosome 18.

**Figure 4 medicina-55-00352-f004:**
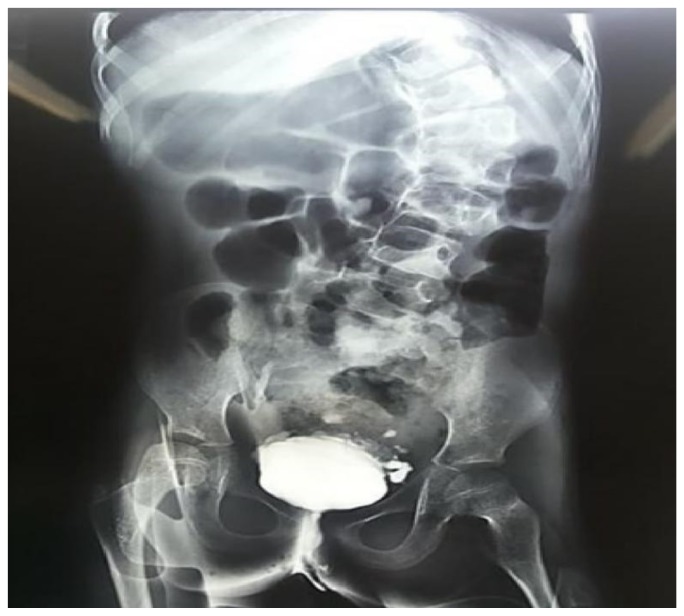
Urinary urethrocystography evidencing marked scoliosis and signs of neurogenic bladder in a 10-year-old girl with trisomy 18 and Dandy-Walker Syndrome.

**Figure 5 medicina-55-00352-f005:**
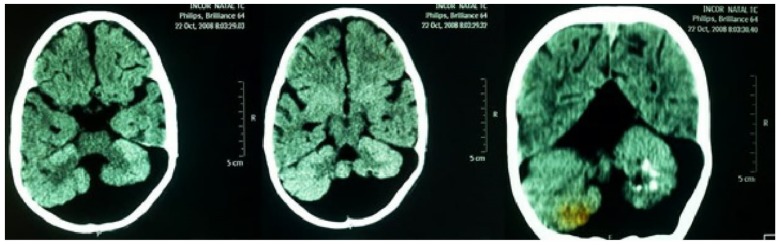
Computed tomography, without contrast of the head, showing massive cystic formation in the posterior fossa inferiorly compressing the occipital parenchyma; hypoplasia of cerebellar vermis with calcifications in the left cerebellar hemisphere. Findings compatible with Dandy-Walker syndrome.

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
