# Peer review of "Long Survival of a Patient with Trisomy 18 and Dandy-Walker Syndrome"

_medicina, 2019, doi:10.3390/medicina55070352_

Reviewer 1 Report

Interesting clinical case of Trisomy 18 given the rarity of cases that reach 12 years of age.

In general, the clinical case is well presented with the clinical symptoms properly described. 

In my opinion, there are 2 areas where I would introduce changes. 

First, I think it would be important to add a more extensive explanation of the difference in sex ratios observed in trisomy 18 given the fact that females normally have a better outcome and the case presented is actually a female.

Second, you mention in the introduction and in the discussion section the reference 8. In the discussion section you write: "Due to the rarity of the association between T18 and DWS, epidemiological data regarding the involvement of these two syndromes are scarce. In a review from 3 series of cases with total of 78 patients with T18, 20 were also affected by the DWS [8]. "

Unfortunately, this is incorrect. If you read carefully that paper, in page 114, it says: "These three reports show that trisomy 18 occurred in 20 of 78 cases of DWS." 

As you can see, it is not the same to observe 20 cases of T18 in 78 cases of DWS than to observe 20 cases of DWS in 78 cases of T18.

This needs to be corrected and more attention needs to be paid when referencing other authors.

Thanks

Author Response

Point 1: I think it would be important to add a more extensive explanation of the difference in sex ratios observed in trisomy 18 given the fact that females normally have a better outcome and the case presented is actually a female.

Response 1: The results presented in the literature on higher survival in females are repeated in several studies. However, the explanation for this finding remains unclear. Due to a correction, we added the description of this knowledge gap in the discussion of our case.

Meyer, R.E.; Liu, G.; Gilboa, S.M.; Ethen, M.K.; Aylsworth, A.R.; Powell, C.M.; Flood, T.J.; Mai, C.T.; Wang, Y.; Canfield, M.A. Survival of children with trisomy 13 and trisomy 18: A multi-state population-based study. American Journal Of Medical Genetics Part A, [s.l.], v. 170, n. 4, p.825-837, 10 dez. 2015. Wiley-Blackwell. 

Rasmussen, SA, Wong, LC, Yang, Q, May, KM, Friedman, JM. 2003. Population‐based analysis of mortality in trisomy 13 and trisomy 18. Pediatrics 111: 777– 784.

Niedrist, D, Riegel, M, Achermann, J, Schinzel, A. 2006. Survival with trisomy 18—Data from Switzerland. Am J Med Genet Part A 140A: 952– 959.

Point 2: You mention in the introduction and in the discussion section the reference 8. In the discussion section you write: "Due to the rarity of the association between T18 and DWS, epidemiological data regarding the involvement of these two syndromes are scarce. In a review from 3 series of cases with total of 78 patients with T18, 20 were also affected by the DWS [8]. "

Unfortunately, this is incorrect. If you read carefully that paper, in page 114, it says: "These three reports show that trisomy 18 occurred in 20 of 78 cases of DWS."

Response 2: As pointed out, there was an error in the interpretation of the reference by the authors, but it has already been corrected. We appreciate the correction.

Reviewer 2 Report

Dear Authors

Reviewer has evaluated a paper by de Souza et al., describing a patient with trisomy 18 and long-term survival after an appropriate surgical intervention and multidisciplinary follow-up. This report would be very valuable because management of children with the condition is still controversy even in developed western countries. Not only large series of children with the condition who had intensive treatment, but also single clinical reports like this paper would be important to share experiences of these approaches. The manuscript is well-written, and only several issues are found to be corrected.

What kind of surgery did she undergo? And how was the postoperative course?

Did she have episodes of urinary tract infection?

Was US in the newborn period available, to demnstrate DWM?

What kind of "multidisciplinary follow-up" did she have? What kind of early intervention did she have?

What kind of discussion did the medical team and her parents have especially in the setting of planning cardiac surgery (performed) and gastrostomy (refused)?

Reviewer also recommends the authors to consider to refer some additional reports from Japan, addressing importance of intensive approaches to children with trisomy 18 as shown below:

Nakamura T, Kawame H, Baba A, Tamura M, Fukushima Y.  Neonatal management of trisomy 18: clinical details of 24 patients receiving intensive treatment. Kosho T, Am J Med Genet A. 2006 May 1;140(9):937-44.

Kosho T, Kuniba H, Tanikawa Y, Hashimoto Y, Sakurai H. Natural history and parental experience of children with trisomy 18 based on a questionnaire given to a Japanese trisomy 18 parental support group.  Am J Med Genet A. 2013 Jul;161A(7):1531-42. doi: 10.1002/ajmg.a.35990. Epub 2013 May 29.

Author Response

Point 1: What kind of surgery did she undergo? And how was the postoperative course?

Response 1: Unfortunately, the patient was operated on a private service outside the university and so the details of the surgery and postoperative are not in our records. In addition, the patient's mother no longer has the discharge summary and the documents provided by the hospital that performed the surgery.

Point 2: Did she have episodes of urinary tract infection?

Response 2: Yes, she did. Regarding the genito-urinary tract, urinary tract (UTI). In renal scintigraphy with DMSA, the presence of horseshoe kidneys and relative renal function of 69% in right kidney and 31% in left kidney. In ultrasonography (USG) of kidneys and urinary tract there were additional findings of hydronephrosis and light hydrotherapy on the right. The urethrocistography revealed the presence of neurogenic bladder associated with

vesico-ureteral reflux to the right of degree II.

Point 3: Was US in the newborn period available, to demnstrate DWM?

Response 3: The diagnoses of DWS as well as of T18 were only after birth.

Point 4: What kind of "multidisciplinary follow-up" did she have? What kind of early intervention did she have?

Response 4: The multidisciplinary follow-up involves the joint action of several medical and non-medical specialties in the integral care of the patient. The professionals involved and their performance are described below:

Cardiologist: preoperative and postoperative follow-up;

Geneticist: Diagnostic;

Nutritionist: Nutritional monitoring;

Neurologist: Follow-up for DWS and seizures;

Gastroenterologist: Follow-up for constipation;

Nephrologist: Follow-up by renal repetition and Nurogenic Bladder.

Point 5: What kind of discussion did the medical team and her parents have especially in the setting of planning cardiac surgery (performed) and gastrostomy (refused)?

Response 5: Cardiac Surgery Planning: It was presented the importance of surgery in improving survival and prognosis of the patient, which was accepted.

Gastrostomy: due to cachexia (Z score -3) gastrostomy was suggested in an attempt to improve nutritional intake. However, it was not accepted by the family due to the risk involved in the procedure, besides being an invasive process which could generate more suffering to the patient.

The authors appreciate the corrections and literature suggestions for the paper.